# Genetically Engineering *Escherichia coli* to Produce Xylitol from Corncob Hydrolysate without Lime Detoxification

**DOI:** 10.3390/molecules28041550

**Published:** 2023-02-06

**Authors:** Xinsong Yuan, Jiyun Cao, Rui Wang, Yu Han, Jinmiao Zhu, Jianping Lin, Lirong Yang, Mianbin Wu

**Affiliations:** 1School of Chemistry and Chemical Engineering, Hefei Normal University, Hefei 230601, China; 2College of Chemical and Biological Engineering, Zhejiang University, Hangzhou 310027, China; 3Ningbo Research Institute, Zhejiang University, Ningbo 315100, China

**Keywords:** CRP, non-detoxification, hemicellulosic hydrolysate, xylitol, *Escherichia coli*

## Abstract

Before fermentation with hemicellulosic hydrolysate as a substrate, it is generally necessary to detoxify the toxic substances that are harmful to microorganism growth. Cyclic AMP receptor protein (CRP) is a global regulator, and mutation of its key sites may have an important impact on *E. coli* virulence tolerance. Using corncob hydrolysate without ion-exchange or lime detoxification as the substrate, shake flask fermentation experiments showed that CRP mutant IS5-dG (I112L, T127G, A144T) produced 18.4 g/L of xylitol within 34 h, and the OD_600_ was 9.7 at 24 h; these values were 41.5% and 21.3% higher than those of the starting strain, IS5-d, respectively. This mutant produced 82 g/L of xylitol from corncob hydrolysate without ion-exchange or lime detoxification during fed-batch fermentation in a 15-L bioreactor, with a productivity of 1.04 g/L/h; these values were 173% and 174% higher than the starting strain, respectively. To our knowledge, this is the highest xylitol concentration and productivity produced by microbial fermentation using completely non-detoxified hemicellulosic hydrolysate as the substrate to date. This study also showed that alkali neutralization, high temperature sterilization, and fermentation of the hydrolysate had important effects on the xylose loss rate and xylitol production.

## 1. Introduction

Converting hemicellulose in plant fiber resources into valuable products, including xylitol through microbial cell factory, is of great significance for the utilization of plant resources, environmental protection, and sustainable development [1,2]. *Escherichia coli* is one of the optimal host organisms for research related to metabolic engineering and synthetic biology because of its metabolic plasticity, its ability to assimilate both hexose and pentose sugars and grow rapidly under commonly used culture conditions, and the availability of substantial biochemical and physiological information and genetic engineering techniques [3]. Metabolic engineering of *E. coli* to produce xylitol has been widely studied by researchers. Nair et al. substantially improved xylose selectivity by a five-point mutation of xylose reductase (XR) [4]. Chin et al. enhanced NADPH regeneration by deleting genes involved in the Embden–Meyerhof–Parnas pathway of glucose metabolism in *E. coli* [5,6]. Su et al. deleted the *xylAB*, *ptsG*, and *ptsF* genes from the *E. coli* W3110 host strain, which greatly improved the xylitol production efficiency with pure xFfylose as a substrate [7]. Su et al. integrated XR into the *E. coli* genome, and the resulting strain, IS5-d, produced 143.8 g/L of xylitol in a 15 L-scale bioreactor using ion-exchange-detoxified corncob hydrolysate as a substrate [8].

Despite these advances, it is still extremely challenging to efficiently produce xylitol from non-detoxified hemicellulosic hydrolysate. The corncob sulfuric acid hydrolysate contains acetic acid, furfural, 5-hydroxymethylfurfural, phenols, sulfate radicals, pectin, pigments, and other toxic substances [9], which can inhibit bacterial cell growth and have serious adverse effects on fermentation production of xylitol by bacterial strains. At present, hydrolysates are generally detoxified by ion exchange or lime treatment. However, ion-exchange detoxification produces a large amount of waste that pollutes the environment and leads to a large loss of xylose. Lime treatment produces some gypsum waste residue, and the evaporator is likely to form a gypsum scale layer during hydrolysate concentration, which is a major disadvantage of lime detoxification.

Constructing a toxicity-tolerant xylitol production strain and developing alternative hydrolysate treatment methods so that the strain can produce xylitol with completely non-detoxified hydrolysate as the substrate should be of great significance for the industrialization of the biological production of xylitol. The biggest difficulty in producing xylitol with completely non-detoxified hydrolysate as the substrate is that the hydrolysate has many inhibitory factors and high toxicity, which make it difficult for bacterial cells to grow. There have been lots of work on metabolic engineering to facilitate the use of non-detoxified hydrolysate or simulated substrate, including screening wild strains [10,11,12] and increasing robustness of the host strains by a genetic engineering strategy including site-directed mutation [13,14] or traditional domestication [15,16].

Cyclic AMP receptor protein (CRP) is a global regulator in *E. coli* that can directly or indirectly regulate the expression of approximately half of all *E. coli* genes. These genes mainly encode DNA transcriptional regulatory proteins, functional enzymes, transporters, membrane proteins, and conserved proteins [14]. Changes in amino acid size, structure, force field, and polarity at key CRP sites may have important effects on protein properties and functions [17]. The 127th amino acid of the CRP protein of *E. coli* W3110 is located in the region near its cAMP binding pocket [18], and its mutations have been reported to alleviate the carbon catabolite repression effect [17,19,20] and improve strain tolerance [18,21]. To improve the robustness of *E. coli* strains through CRP mutation so that it can produce xylitol with the completely non-detoxified hemicellulose hydrolysate as the substrate, avoiding the complicated detoxification process and the generation of pollutants, should be of great significance to the industrialization of the biological production of xylitol.

On the basis of the previously constructed *E. coli* strain IS5-dI [22], we generated a series of strains via site-directed mutagenesis targeting Ile^127^ of CRP. Then, hydrolysate was used as the substrate without ion exchange or lime treatment; the strains were screened through shaking flask fermentation experiments, and the most toxicity-tolerant strain was used as the xylitol-producing strain. The influence of various factors on xylitol production, including pH value of the hydrolysate, evaporation concentration, high-temperature sterilization, and fermentation process, was studied. Finally, the toxicity-tolerant strain IS5-dG was used as the host strain for xylitol production by fermentation in a 15-L bioreactor with completely non-detoxified hydrolysate as the substrate.

## 2. Results and Discussion

### 2.1. Relationship between Xylose Loss Rate and pH of Neutralized Hydrolysate

The effects of pH on the xylose loss rate during neutralization, concentration, and sterilization (115 °C, 0.5 h) are shown in Table 1. When the neutralization pH value was 4.0, the xylose loss rates of neutralization, concentration, and sterilization and the total xylose loss rate were 0.7, 2.2, 4.1, and 6.9, respectively. When the pH did not exceed 4.0, the loss rate of xylose during concentration and sterilization was small. When the pH value exceeded 4.5, the loss rate of xylose during concentration and sterilization sharply increased with elevated pH value. When the pH was 7.5, the total loss rate of xylose was about 30.9%.

The chromaticity of the hydrolysate obtained from neutralization to different pH values is shown in Figure 1. The pH did not exceed 4.0 and the hydrolysate was mainly light brown. The color of the hydrolysate at pH 4.5 started to darken, and it became darker as the pH increased. When the pH value was high, during the concentration process, the aldol condensation reaction of the hydroxyl and aldehyde groups of xylose was alkali catalyzed to generate a dark resinous jelly-like substance; this resulted in the loss of xylose. Under high-temperature sterilization, in addition to aldol condensation, the aldehyde–ammonia condensation reaction of amino groups and aldehyde groups also occurred, which resulted in the formation of black gum and loss of xylose. Because corncob generally contains 2–6% crude protein, we speculate that the ammonia in this acetaldehyde condensation reaction mainly comes from this protein. Therefore, to reduce xylose loss, the pH after neutralization should be controlled so that it does not exceed 4.0.

### 2.2. Influence of pH Value on Removal Rate of Acetic Acid during Concentration

Table 2 shows the HPLC detection index of a batch of typical corncob hydrolysate provided by Huakang Co (Quzhou, China). The acetic acid concentration exceeded 8 g/L, which may be the main inhibitory factor for fermentation of the *E. coli* strains in this study.

The volatile components, including furfural and acetic acid, were mainly removed by vacuum concentration. The hydrolysate was neutralized by NaOH solution, and the pH value after neutralization had a significant effect on the acetic acid removal rate during concentration (Table 3). When the pH value did not exceed 4.0, most of the acetic acid could be removed by vacuum concentration. When the pH value was 5.5, the acetic acid removal rate during concentration was only 42%. According to the calculation of the dissociation constant of acetic acid, when the pH value does not exceed 4.0, the acetate group mainly exists in the form of free acetic acid. Additionally, when the pH value exceeded 5.5, most of the acetate group existed in the form of acetate. Free acetic acid is a volatile substance and can be easily removed by evaporation, but acetate is not volatile and cannot be removed by evaporation.

### 2.3. Effect of Sterilization Temperature on Loss Rate of Xylose

We found that the commonly used high-temperature (115 °C or 121 °C) sterilization caused some loss of xylose. Table 4 shows the influence of sterilization temperature on the xylose loss rate in the concentrated hydrolysate with a pH value of 4.0. After sterilization at 100 °C, 110 °C, 115 °C, and 121 °C for 0.5 h, the xylose loss rates during sterilization were 1.5%, 3.0%, 4.1%, and 10.8%, respectively. Moreover, no bacterial growth was found in the hydrolysate sterilized at 100 °C for 0.5 h after 10 streaks of antibiotic-free LB solid medium and 10 streaks of LB liquid medium and incubation for 24 h. This indicates that the concentrated hydrolysate was completely sterilized by sterilization at 100 °C for 0.5 h in this study. Therefore, we chose 100 °C for 0.5 h as the sterilization conditions for the concentrated hydrolysate.

### 2.4. Screening of Toxicity-Tolerant Strains by Shake Flask Fermentation

Figure 2 shows the results of shake flask fermentation using the starting strain IS5-d (a) and various strains (b–j) that were constructed via site-directed mutagenesis for xylitol production from corncob hydrolysate without lime treatment. For the starting strain, IS5-d, the OD_600_ peaked at 8.0 at 24 h, and the maximum xylitol concentration was 13 g/L at 34 h. For IS5-dI, IS5-dL, IS5-dA, IS5-dF, IS5-dM, IS5-dV, IS5-dP, and IS5-dW (Figure 2b–i, respectively), the maximum OD_600_ did not exceed 5.0 and less than 7 g/L of xylitol was produced. These findings demonstrate that the growth of these mutants was strongly inhibited by the toxic substances of the hydrolysate; therefore, they are not suitable for use as xylitol-producing strains.

Strain IS5-dG (Figure 2j) produced 18.4 g/L of xylitol within 34 h and the OD_600_ was 9.7 at 24 h, which were 41.5% and 21.3% higher than those of the starting strain, respectively. The data of the three parallel experiments demonstrated that the xylitol titers achieved by these two strains very significantly differed (*p* < 0.01). Therefore, IS5-dG was the only positive mutant whose toxicity tolerance was significantly improved, and it is the most promising strain for xylitol production from the non-detoxified hydrolysate.

The substrate used in this study is the hydrolysate without lime treatment, while the substrate in our previous work is the hydrolysate after lime treatment. Except that, the experimental method of this work is very similar to that of the literature [22]. The results presented in Figure 2 of this study look similar to that in the previous paper; however, in this work, the strain growth level and xylitol concentration are lower than those in the literature. The results showed that the hydrolysate without lime treatment had a stronger inhibitory effect on cell growth and xylitol production.

### 2.5. Xylitol Production from Corncob Hydrolysate without Lime Treatment by IS5-dG

The IS5-dG strain produced 82 g/L of xylitol from corncob hydrolysate without lime detoxification after a 79-h fed-batch fermentation in a 15-L bioreactor. The bacterial growth and sugar concentrations are presented in Figure 3a. The xylitol yield per xylose was 0.95 g/g and xylitol productivity was 1.04 g/L/h. Additionally, 5 g/L of residual xylose was detected, but no other residual sugars were detected in the fermentation broth. In our previous work, with the lime-detoxified hydrolysate as the substrate, xylitol concentration and productivity were higher at 137 g/L and 1.76 g/L/h, respectively. These results further indicate that there are some toxic substances in the hydrolysate that have adverse effects on xylitol production, which can be effectively improved by lime treatment. However, the lime treatment process produces lots of pollution, while the purpose of this study is to explore a xylitol production process that avoids pollution by using lime treatment, in order to provide a more green strategy for xylitol production.

The control strain IS5-d produced 30 g/L of xylitol after a 79-h fermentation (Figure 3b), and xylitol productivity was 0.38 g/L/h. The residual xylose, glucose, and arabinose concentrations were 53 g/L, 1 g/L, and 1 g/L, respectively, and the calculated xylitol yield per xylose was 0.35 g/g. The maximum OD_600_ value of IS5-d was only 31, which indicated that the toxic substances in the hydrolysate significantly inhibited strain growth.

The xylitol concentration and productivity produced by IS5-dG were 173% and 174% higher than those of IS5-d, respectively.

The suitable neutralization pH for this work is 4.0, and a pH value that is too low is unfavorable for producing xylitol during fermentation. At a neutralized pH of 3.0, the xylitol concentration produced under the same conditions was 55 g/L. We speculate that this occurred for the following reasons. First, ammonia solution was used as an alkali source to adjust the pH value of the acidic fermentation broth, and it was also used as an inorganic nitrogen source for bacterial growth. If the pH of the hydrolysate was too low, the pH needed to be neutralized with a large amount of ammonia solution; this resulted in excessive ammonia concentration in the fermentation broth, which may have severe adverse effects on bacterial growth and xylitol production.

### 2.6. Effects of Sulfate Ion Inhibition on Strain Growth and Xylitol Production

To study the effect of sulfate ion inhibition on strain growth and xylitol production, we prepared different concentrations of ammonium sulfate aqueous solution, which was then added to the shake flask fermentation medium. Subsequently, the shake flask fermentation experiment was carried out (as described in Section 2.4). The OD_600_ and xylitol concentration of the broth fermented in shake flasks for 20 h with pure sugar as the substrate were used as the main indicators, and different ammonium sulfate concentrations (g/L: 0, 10, 20, 30, 40, 50, 60, 70) were investigated. The inhibition of xylitol production by IS5-dG (Figure 4) was used to simulate the effect of sulfate on xylitol production under fermenter fermentation conditions. Without ammonium sulfate, the OD_600_ of the bacteria was 11, and 12 g/L of xylitol was produced. With increased sulfate concentration, the OD_600_ of the bacteria gradually decreased, and the xylitol concentration decreased. There was a serious inhibitory effect of 20 g/L (NH_4_)_2_SO_4_ on bacterial growth and xylitol production. When the added (NH_4_)_2_SO_4_ concentration reached 40 g/L, bacterial growth was hindered, and xylitol production was also severely inhibited.

The xylose concentration in the original hydrolysate (sulfate ion concentration of about 50 mM) was approximately 50 g/L. Under fermentation conditions, the total xylose concentration in the bioreactor usually reached more than 100 g/L, and the (NH_4_)_2_SO_4_ concentration exceeded 13 g/L. Figure 4 shows that the OD_600_ and xylitol concentrations were 8.6 and 8.8 g/L, respectively, which were 21.8% and 26.7% lower than those without sulfate ions, indicating that the high sulfate concentration in the hydrolysate had adverse effects on both bacterial growth and xylitol production. Regarding the inhibition mechanism of sulfate ions, previous studies [23,24] reported that high osmotic pressure produced by high concentrations of salt might result in reduced cell growth. However, the specific inhibition mechanism is still unclear and warrants further investigation.

### 2.7. Effect of Glucose Concentration on Xylitol Productivity

Feeding mode and glucose concentration control are very important for xylitol production by *E. coli* fermentation. First, when the glucose concentration is too high, the cells absorb glucose too quickly, the oxidation is incomplete, the metabolism is incomplete, and it is easy to produce organic acids such as acetic acid, which can inhibit cell growth. Second, a high glucose concentration inhibits CRP expression [25]. The promoter of the xylose operon is CRP-dependent, and its activity is proportional to the concentration of intracellular CRP [26]; therefore, a high glucose concentration will inhibit xylose transport. Finally, a high glucose concentration will produce feedback inhibition of xylose conversion. Rosa et al. [27] found that, when glucose concentration exceeded 5 g/L, XR activity and xylitol production were reduced. Walther et al. [28] also found that the presence of glucose inhibited XR activity, thereby reducing xylitol productivity. Therefore, glucose should be supplemented by continuous feeding and its concentration should be controlled so that it does not exceed 10 g/L (preferably, it should not exceed 5 g/L).

Compared with the ion-exchange detoxification process, lime detoxification is simple and inexpensive, and almost no waste water is produced. During the neutralization of lime, lime and sulfate ions react to form calcium sulfate, which exists in the form of colloid under certain conditions and can absorb inhibitors such as acetic acid, furfural, pectin, and pigment in the hydrolysate. From the perspective of industrialization, lime detoxification has great advantages over ion exchange. However, there are still some problems in the lime treatment process. This process produces some solid waste, and the evaporator easily forms a scale layer, which shortens its service life. This research presents a systematic study on the neutralization, evaporative concentration, sterilization, and fermentation of corncob hydrolysate. On this basis, the IS5-dG strain produced 82 g/L of xylitol from the hydrolysate without lime treatment in a 15-L bioreactor fermentation experiment. To our knowledge, this is the highest concentration of xylitol produced by microbial fermentation in a completely non-detoxified hydrolysate.

## 3. Materials and Methods

### 3.1. Strains, Plasmids, Primers, Reagents, and Culture Medium

The bacterial strains and plasmids used in this study are listed in Appendix A. The primer pairs used in this study are listed in Appendix A. PrimeSTAR Max DNA Polymerase, Gel DNA Purification Kits, and Plasmid Midi Kits were obtained from Axygen Scientific, Inc. (Hangzhou, China). D-xylose, 4-morpholinopropanesulfonic acid, spectinomycin, and kanamycin were purchased from Genview Scientific, Inc. (Carlsbad, CA, USA). Tryptone and yeast extract were obtained from Oxoid (Basingstoke, UK). Primers were synthesized and sequencing reactions were completed by Sangon Biotech (Shanghai, China). Corncob hydrolysates (xylose concentration, 57 g/L; SO_4_^2−^ molar concentration, 0.05 M; pH 1.6) were provided by Huakang Pharmaceutical Co., Ltd. (Zhejiang, China). The method of generating corncob hydrolysates mainly includes a series of physicochemical and enzymatic treatment processes. The corncob powders were firstly treated with pectinase, and then reacted with dilute sulfuric acid (0.6~1.0%) at 125 °C for 50 min. Hemicellulose was almost completely hydrolyzed to xylose, and cellulose and lignin remained unchanged as solid residues, which were then separated by a plate and frame filter. The lignin content in corncob and acidolysis residue is about 20% and 35%, respectively. The hydrolysate contains almost no lignin. The detailed production process of hydrolysate is described in the patent of Huakang Company [29]. Dry corn steep liquor powders were provided by Haizheng Pharmaceutical Co. (Zhejiang, China). All compounds were of reagent grade quality or higher.

*Escherichia coli* strains were cultured in Luria–Bertani (LB) medium (per liter: 10 g tryptone, 5 g yeast extract, and 10 g NaCl) at 37 °C with vigorous shaking (200 rpm). The primary seed medium was LB medium. The secondary seed medium included the following (per L): 7.5 g yeast extract, 7.5 g peptone, 15 g glucose, and 10 g NaCl. The glucose solution was separately sterilized. Shake flask fermentation medium included the following (per L): 6 g Na_2_HPO_4_·12H_2_O, 3 g KH_2_PO_4_, 1 g NH_4_Cl, 0.5 g NaCl, 1 mM MgSO_4_, 1 mM CaCl_2_, 5 g yeast powder, 1 mL 1000× Trace Metals, and 50 mM 4-morpholinopropanesulfonic acid (pH 7.4, for pH stabilization). For bioreactor cultures, bacteria were primarily grown in modified M9 minimal medium (per L: 15 g glucose, 15 g yeast extract, 9 g Na_2_HPO_4_·12H_2_O, 3 g KH_2_PO_4_, 1 g NH_4_Cl, and 0.5 g NaCl), and corn steep liquor solution was added as a nitrogen source in subsequent fermentation.

### 3.2. Hydrolysate Neutralization, Evaporative Concentration, and Sterilization

The corncob hydrolysate was neutralized to pH 2.5, 3.0, 3.5, 4.0, 4.5, 5.0, and 5.5 with 1 M NaOH, and then concentrated to a xylose concentration of 500 g/L using a rotary evaporator (RE-1002 10 L type, Hangzhou Huichuang Instrument Co., Ltd., Zhejiang, China). The concentrated hydrolysate was sterilized in an autoclave at 115 °C for 0.5 h, and the influence of the pH value (NaOH dosage) on the loss rate of xylose during neutralization, evaporation concentration, and sterilization was investigated. Furthermore, the effects of different sterilization temperatures (100 °C, 110 °C, 115 °C, and 121 °C) on the loss rate of xylose in the hydrolysate were investigated.

### 3.3. Genetic Methods

A series of genetically engineered strains were constructed with a CRISPR/Cas9 gene-editing system [30] and described in our previous report [22]. Using the xylitol production strain IS5-d, which was previously constructed in our laboratory, as the starting strain, IS5-dI was obtained by mutation of *crp* (I112L, T127I, A144T). On this basis, the 127th amino acid of IS5-dI was further mutated into eight other kinds of non-polar amino acids, L, A, F, M, V, P, W, and G, to obtain eight xylitol-producing strains: IS5-dL, IS5-dA, IS5-dF, IS5-dM, IS5-dV, IS5-dP, IS5-dW, and IS5-dG (see Appendix A for details).

### 3.4. Shake Flask Fermentation

A seed culture was prepared by inoculating 5 mL medium (15 × 150 mm tube) with a single colony from a fresh LB agar plate. The shake flask fermentation experiment was conducted with sterile 250-mL Erlenmeyer flasks containing 45 mL medium. The medium was inoculated with approximately 1 mL overnight culture and then incubated at 37 °C with vigorous shaking (200 rpm). After approximately 4 h, a 5-mL solution containing 200 g/L xylose and 100 g/L glucose (or hemicellulosic sugars supplemented with an appropriate amount of glucose) was added to the culture. The flasks were then maintained at 30 °C with shaking (200 rpm), and the cultures were sampled at regular intervals and then analyzed.

### 3.5. Bioreactor Fermentation

The shake flask fermentation experiment showed that the IS5-dG strain was the only toxicity-tolerant forward mutant, and it grew normally with hydrolysate without lime treatment as the substrate. Therefore, IS5-dG was selected to analyze xylitol production from condensed corncob hydrolysates via fed-batch fermentation in a 15-L bioreactor (Guoqiang Biochemical Engineering Equipment Co., Ltd., Shanghai, China).

The bioreactor initially contained 6.5 L modified M9 minimal medium, 600 mL IS5-dG seed culture, and 24 g/L corn steep liquor. The initial pH value was set to 6.8, which was adjusted with ammonium hydroxide. Once the pH value was lower than 6.7, the ammonium hydroxide was automatically replenished by the peristaltic pump. The dissolved oxygen level was maintained at 20–30% by controlling the agitation speed, and the temperature was maintained at 30 °C. When the OD_600_ approached 20 (after about 10 h), approximately 0.8 L concentrated corncob hydrolysate (final xylose concentration of about 53 g/L) and 0.2 L corn steep liquor (final concentration of 23 g/L) were added by a peristaltic pump. Because the pH value of the hydrolysate was very low, it was added slowly (8–10 min) so that ammonia hydroxide could be replenished in time to avoid inhibition of bacterial growth due to the low pH value. An appropriate amount of glucose was added to the bioreactor in a fed-batch manner, and the glucose concentration was controlled so that it did not exceed 10 g/L. When the xylose concentration decreased to below 25 g/L, an appropriate amount of corncob hydrolysate and a corn steep liquor solution were added to the bioreactor. The total xylose concentration added to the bioreactor was about 86 g/L, and the overall fermentation volume was approximately 10.5 L. Cultures were sampled at regular intervals and then analyzed.

As a control, xylitol production by IS5-d from the hydrolysate without lime detoxification was evaluated using similar fermentation conditions to those used for IS5-dG.

### 3.6. High-Performance Liquid Chromatography Analysis

Xylitol, xylose, arabinose, and glucose concentrations were quantified using a Shimadzu LC-20AT high-performance liquid chromatography (HPLC) system equipped with a refractive index detector (RID-20A) (Shimadzu, Japan) and a BioRad Aminex HPX-87H column (300 × 7.8 mm) (Bio-Rad, Hercules, CA, USA). Additionally, 5 mmol/L H_2_SO_4_ was used as the mobile phase at a column temperature of 65 °C at a flow rate of 0.6 mL/min.

### 3.7. Statistical Analysis

All data points reported are the average of three parallel experiments (the individual plots are shown in the Appendix A), except for the 15-L fermentation experiment, which presented the data of one typical batch of three parallel experiments (the other two sets of plots are shown in Appendix A). The error bars in the figures represent standard deviations of the data from the three parallel experiments. The *p*-value was calculated based on xylitol titers of the three parallel bioreactor fermentation experiments for strains IS5-dG and IS5-d. The difference was considered significant if *p* < 0.05 and very significant if *p* < 0.01.

## 4. Conclusions

A CRP mutant with significantly improved robustness, IS5-dG, was constructed from the xylitol-producing *E. coli* strain IS5-d. IS5-dG can produce xylitol with corncob hydrolysate without ion-exchange or lime detoxification as the substrate. In the shake flask fermentation experiment, IS5-dG produced 18.4 g/L of xylitol within 34 h, and the OD_600_ was 9.7 at 24 h; these values were 41.5% and 21.3% higher than those of the starting strain, IS5-d, respectively. In a fed-batch fermentation experiment involving a 15-L bioreactor at 30 °C for 79 h, IS5-dG produced 82 g/L of xylitol from corncob hydrolysate without lime treatment and had a productivity of 1.04 g/L/h; these values were 173% and 174% higher than those of the starting strain. Regarding the concentration and sterilization of the hydrolysate, to achieve a low xylose loss rate, the most suitable conditions included a pH value of 4.0 for NaOH neutralization of the hydrolysate and a sterilization temperature of 100 °C. The feeding strategy was coupled with the slow addition of the hydrolysate by a peristaltic pump and the immediate addition of ammonia solution to adjust the pH value to about 6.8. Glucose should be supplemented by continuous feeding and its concentration controlled so that it does not exceed 10 g/L.

## Figures and Tables

**Figure 1 molecules-28-01550-f001:**
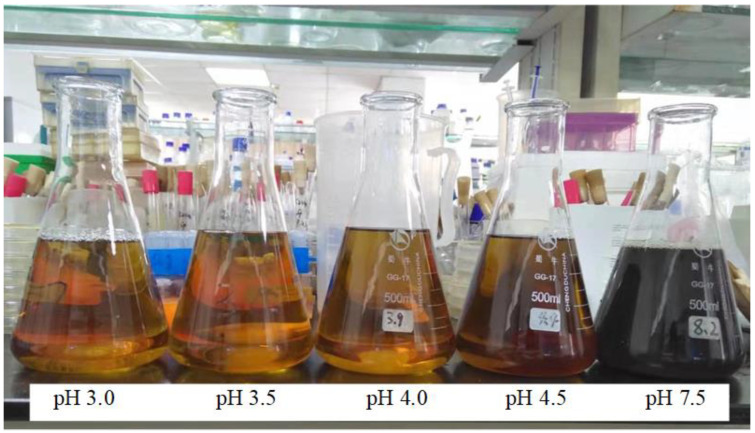
Effect of different pH values on hydrolysate color.

**Figure 2 molecules-28-01550-f002:**
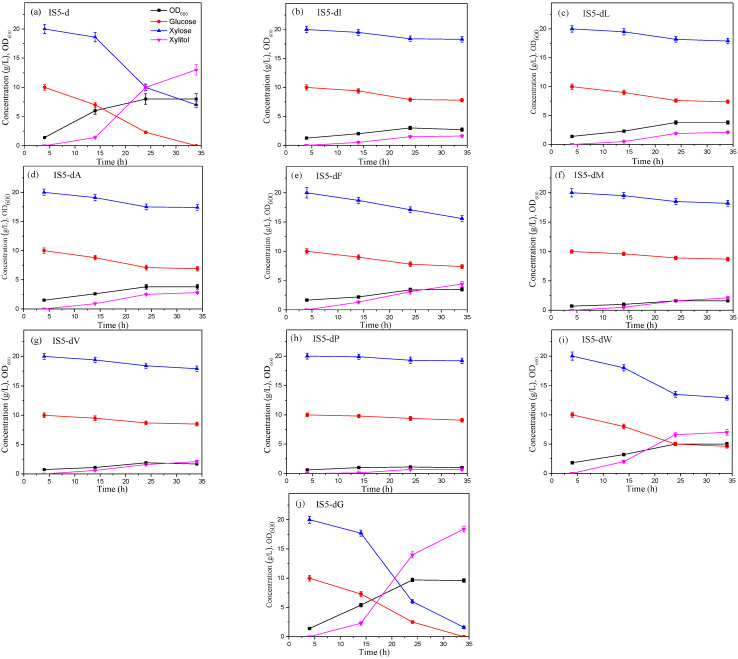
Shake flask fermentation by the starting strain IS5-d (**a**) and various strains (**b**–**j**) constructed via site-directed mutagenesis for xylitol production from corncob hydrolysate without lime treatment. (**a**) For the starting strain, IS5-d, the OD600 peaked at 9.4 at 24 h, and the maximum xylitol concentration was 15 g/L at 34 h. For (**b**) IS5-dI, (**c**) IS5-dL, (**d**) IS5-dA, (**e**) IS5-dF, (**f**) IS5-dM, (**g**) IS5-dV, and (**h**) IS5-dP, the maximum OD600 did not exceed 4.0 and less than 4.7 g/L of xylose was converted to xylitol. Additionally, most of the glucose was not consumed. (**i**) Strain IS5-dW consumed 7 g/L of glucose and converted 12 g/L of xylose to xylitol, with a maximum OD600 of 6.6. (**j**) Strain IS5-dG produced 19 g/L of xylitol within 34 h, with no residual glucose detected, and the OD600 was 11.6 at 24 h. Individual plots are presented in Appendix A.

**Figure 3 molecules-28-01550-f003:**
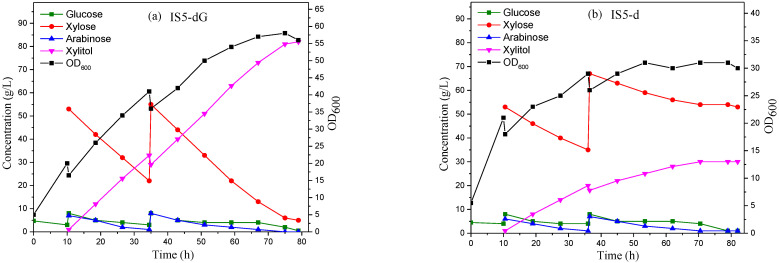
Bacterial growth and sugar concentrations of the fed-batch fermentation by IS5-dG (**a**) and IS5-d ((**b**), as a control) to produce xylitol from corncob hydrolysate without lime treatment. The plots of the two sets of repeated experiments are presented in Appendix A.

**Figure 4 molecules-28-01550-f004:**
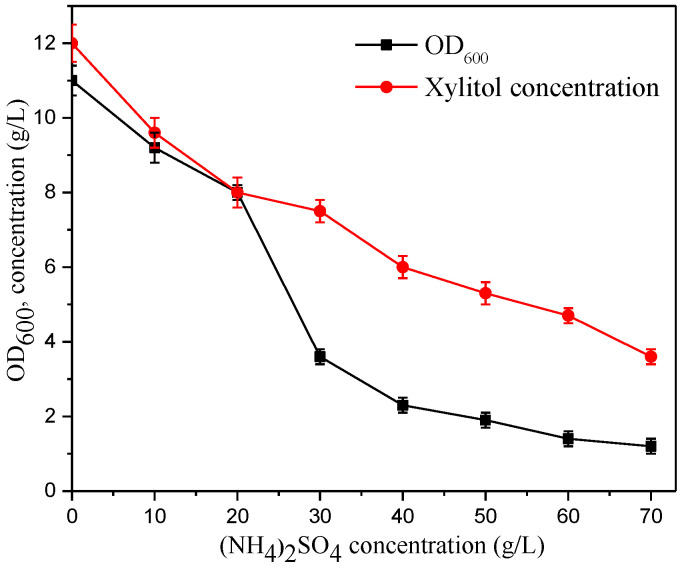
Effects of SO_4_^2−^ inhibition on IS5-dG strain growth and xylitol production. Individual plots are presented in Appendix A.

**Table 1 molecules-28-01550-t001:** Effect of different pH values on xylose loss during concentration and sterilization.

pH	2.5	3.0	3.5	4.0	4.5	5.0	5.5	7.5
Xylose loss rate after neutralization (%)	0	0	0	0.7 ± 0.1	1.2 ± 0.1	1.7 ± 0.2	2.7 ± 0.2	3.3 ± 0.2
Xylose loss rate after concentration (%)	0	0	1.3 ± 0.1	2.1 ± 0.2	3.3 ± 0.3	5.1 ± 0.3	9.4 ± 0.4	10.0 ± 0.5
Xylose loss rate after sterilization (%)	0	0.7 ± 0.1	1.9 ± 0.2	4.1 ± 0.2	5.8 ± 0.3	7.7 ± 0.4	16.7 ± 1.1	17.6 ± 1.3
Total xylose loss rate (%)	0	0.7 ± 0.1	3.2 ± 0.3	6.9 ± 0.5	10.3 ± 0.7	14.5 ± 0.9	28.8 ± 1.7	30.9 ± 2.0

**Table 2 molecules-28-01550-t002:** Main HPLC detection indexes of the hydrolysate provided by Huakang Co.

Compound	Retention Time (min)	Peak Area	Peak Area Ratio (%)	Concentration (g/L)
Glucose	8.074	118,743.62	5.68	4.23
Xylose	8.644	1,590,795.26	76.03	57.06
Arabinose	9.398	139,295.26	6.66	1.43
HMF	12.106	6003.76	0.29	<1
Furfural	12.542	1296.76	0.06	<1
Acetic acid	13.660	107,948.56	5.16	8.05

**Table 3 molecules-28-01550-t003:** Effect of different pH values on the acetic acid removal rate.

pH Value	2.5	3.0	3.5	4.0	4.5	5.0	5.5
Acetic acid removal rate (%)	95 ± 3.0	93 ± 2.6	91 ± 2.8	88 ± 2.5	74 ± 2.2	61 ± 2.0	42 ± 1.6

**Table 4 molecules-28-01550-t004:** Effect of different sterilization temperatures on the xylose loss rate of hydrolysate.

Sterilization Temperature (°C)	100	110	115	121
Xylose loss rate (%)	1.5 ± 0.2	3.0 ± 0.3	4.1 ± 0.3	10.8 ± 0.5

## Data Availability

The data presented in this study are available on request from the corresponding author.

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
