# Peer review of "Genetically Engineering *Escherichia coli* to Produce Xylitol from Corncob Hydrolysate without Lime Detoxification"

_molecules, 2023, doi:10.3390/molecules28041550_

Round 1
Reviewer 1 Report
The paper describes a xylose to xylitol fermentation using corn-cob hydrolysate as the feedstock and an engineered strain. The results presented in Figure 2 of this paper look very similar to what was published as Figure 6 in a previous paper by author M. Wu: reference 16: AMB 2020, 104, 2039. If this is published data, the authors should cite the previous paper in the Figure 2 caption.
Moreover, the definition of non-detoxified hydrolysate is a little misleading, since some some pre-treatment steps were performed to remove the acetic acid and volatiles before fermentation.
The authors should comment on why xylitol productivity is higher (1.76 g/l/h) in the 2020 reference than this paper (1.04 g/l/h) also using non-detoxified hydrolysate for IS5-dG. How do the two hydrolysates differ?
Author Response
- The paper describes a xylose to xylitol fermentation using corn-cob hydrolysate as the feedstock and an engineered strain. The results presented in Figure 2 of this paper look very similar to what was published as Figure 6 in a previous paper by author M. Wu: reference 16: AMB 2020, 104, 2039. If this is published data, the authors should cite the previous paper in the Figure 2 caption.
Response: Thank you for your suggestion. We have made relative supplements. The substrate used in this study is the hydrolysate without lime treatment, while the substrate in our previous work is the hydrolysate after lime treatment. Beyond that, the experimental method of this work is very similar to that of the literature [23]. The results presented in Figure 2 of this study look similar to Figure 6 in the previous paper [23], however, in this work, the strain growth level and xylitol concentration are lower than those in the literature. The results showed that the hydrolysate without lime treatment had a stronger inhibitory effect on cell growth and xylitol production.
- Moreover, the definition of non-detoxified hydrolysate is a little misleading, since some pre-treatment steps were performed to remove the acetic acid and volatiles before fermentation.
Response: Thank you for your constructive criticism. We have revised “non-detoxified hydrolysate” to “hydrolysate without lime detoxification” in title and main body of the manuscript.
- The authors should comment on why xylitol productivity is higher (1.76 g/l/h) in the 2020 reference than this paper (1.04 g/l/h) also using non-detoxified hydrolysate for IS5-dG. How do the two hydrolysates differ?
Response: Thank you for your comments. We have made corresponding explanations as follows. In our previous work, with the lime detoxified hydrolysate as the substrate, xylitol concentration and productivity were higher as 137 g/L and 1.76 g/L/h, respectively. These results further indicate that there are some toxic substances in the hydrolysate that have adverse effects on xylitol production, which can be effectively improved by lime treatment. However, lime treatment process produces lots of pollution, while the purpose of this study is to explore a xylitol production process that avoids pollution by using lime treatment, in order to provide a more green strategy for xylitol production.
Reviewer 2 Report
The manuscript describes the production of xylitol from a non-detoxified hydrolysate stream.
1. Authors are suggested to include appropriate citations in the introduction from other's work on metabolic engineering to facilitate the use of non-detoxified hydrolysates.
2. Please include the xylose loss in Table 1 for pH 7.5 as well to help readers.
3. Authors described the condensation reaction between aldehyde of xylose and ammonia as one of the reasons for the loss of xylose with increasing pH. Please elaborate the source of ammonia (in that quantity) especially when pH adjustment was done with NaOH (as described in the experimental section). The experimental section must be written clearly - to help readers understand it and reproduce the work.
4. Would authors not consider the presence of lignin as a reason for darkening of the hydrolysates with increasing pH? Several studies in the literature have described and demonstrated such effects.
5. Please also include the method to generate corncob hydrolysate. Please explain the lower concentrations of glucose in these hydrolysates.
6. Please include a motivation to describe why detoxification was attempted earlier in the manuscript when the intent was to utilize non-detoxified hydrolysates.
Author Response
- Authors are suggested to include appropriate citations in the introduction from other's work on metabolic engineering to facilitate the use of non-detoxified hydrolysates.
Response: Thank you for your suggestion. In another reviwer's comment, the definition of non-detoxified hydrolysate is a little misleading, since some pre-treatment steps were performed to remove the acetic acid and volatiles before fermentation. We have revised “non-detoxified hydrolysate” to “hydrolysate without ion-exchange or lime detoxification”.
Therefore, we have made relative supplements as follows. There have been lots of work on metabolic engineering to facilitate the use of non-detoxified hydrolysate or simulated substrate, including screening wild strains [*1-*3], and increasing robustness of the host strains by genetic engineering strategy including site-directed mutation [*4,*5] or traditional domestication [*6,*7].
[*1] Almeida, J.R.M.; Karhumaa, K.; Bengtsson, O.; Gorwa-Grauslund M.F. Screening of Saccharomyces cerevisiae strains with respect to anaerobic growth in non-detoxified lignocellulose hydrolysate. Bioresour. Technol., 2009, 100, 3674–3677; DOI10.1016/j.biortech.2009.02.057.
[*2] Zhao, M.L.; Shi, D.C.; Lu, X.Y.; Zong, H.; Zhuge, B.; Ji, H. Ethanol fermentation from non-detoxified lignocellulose hydrolysate by a multi-stress tolerant yeast Candida glycerinogenes mutant. Bioresour. Technol., 2019, 273, 634-640; DOI10.1016/j.biortech.2018.11.053.
[*3] Bazoti, S.F.; Golunski, S.; Siqueira, D.P.; Scapini, T.; Barrilli, E.T.; Mayer, D.A.; Barros, K.O.; Rosa, C.A.; Stambuk, B.U.; Alves Jr., S.L.; Valério, A.; Oliveira, D.D.; Treichel, H. Second-generation ethanol from non-detoxified sugarcane hydrolysate by a rotting wood isolated yeast strain. Bioresour. Technol., 2017, 244, 582-587; DOI10.1016/j.biortech.2017.08.007.
[*4] Geng, H.; Jiang, R. cAMP receptor protein (CRP)-mediated resistance/tolerance in bacteria: mechanism and utilization in biotechnology. Appl. Microbiol. Biotechnol. 2015, 99, 4533–4543; DOI 10.1007/s00253-015-6587-0.
[*5] Ma, H.W.; Kumar, B.; Ditges, U, Gunzer, F.; Buer, J.; Zeng, A.P. An extended transcriptional regulatory network of Escherichia coli and analysis of its hierarchical structure and network motifs. Nucleic Acids Res. 2004, 32, 6643–6649; DOI:10.1093/nar/gkh1009
[*6] Kim, S.J.; Yoon, J.; Im, D.K.; Kim, Y.H.; Oh, M.K. Adaptively evolved Escherichia coli for improved ability of formate utilization as a carbon source in sugar-free conditions. Biotechnol. Biofuels, 2019, 12 (1); DOI.org/10.1186/s13068-019-1547-z.
[*7] Yuan, Q.P.; Fan X.G.; Wang X.L.; Wang, L.; Zhu, X.T. A method for preparing high value-added sugar alcohols from lignocellulosic materials with high efficiency. Chinese patent. 2013, CN102876732A.
- Please include the xylose loss in Table 1 for pH 7.5 as well to help readers.
Response: Thank you for your suggestion. We have made corresponding supplements.
- Authors described the condensation reaction between aldehyde of xylose and ammonia as one of the reasons for the loss of xylose with increasing pH. Please elaborate the source of ammonia (in that quantity) especially when pH adjustment was done with NaOH (as described in the experimental section). The experimental section must be written clearly - to help readers understand it and reproduce the work.
Response: Thank you for your constructive suggestion. Because corncob generally contains 2%-6% crude protein, we speculate that the ammonia in this acetaldehyde condensation reaction mainly comes from this protein. We have made corresponding supplement.
- Would authors not consider the presence of lignin as a reason for darkening of the hydrolysates with increasing pH? Several studies in the literature have described and demonstrated such effects.
Response: After special physicochemical and enzymatic treatment of corncob by Huakang Company, we speculate that the blackening caused by lignin may exist, but the impact is limited. However, the exact extent of this impact still needs further study.
- Please also include the method to generate corncob hydrolysate. Please explain the lower concentrations of glucose in these hydrolysates.
Response: As regard to the method to generate corncob hydrolysate, we have made relative supplement in Part 3: Materials and Methods. The method of generating corncob hydrolysate mainly includes a series of physicochemical and enzymatic treatment processes. For details, please refer to Huakang's patent [*8]. For the lower concentrations of glucose in these hydrolysate, we are not very clear about this detail because the enterprise keeps it confidential. The relevant data of corncob hydrolysate generated by this method is relatively stable, and we always provide these original data objectively. As shown in Table 5.5 in Figure 1, please refer to Dr. Yuan Xinsong's graduation thesis for details [*9].
[*8] Jiang, S.T.; Chen, D.S.; Li, M.; Liao, C.J.; Mao, B.X.; Zheng, Y.; Zhang, J.J.; Li, S.; Wang, X.X. The invention relates to a method for preparing xylose solution from corncob. Chinese patent. 2018, CN107893131A.
[*9] Yuan, X.S. Further improvement of E. coli IS5-d for xylitol production via CRP mutation and enhancement of NADPH regeneration (in Chinese). Thesis for doctor's degree of Zhejiang University, PR China, 2020.
- Please include a motivation to describe why detoxification was attempted earlier in the manuscript when the intent was to utilize non-detoxified hydrolysates.
Response: From the perspective of the industrialization of xylitol production by biological method, we pursue not only the high concentration and high productivity of xylitol, but also the green production process. However, our goal is gradual and cannot be achieved overnight. In our previous work [*10], with the lime detoxified hydrolysate as the substrate, xylitol concentration and productivity were higher as 137 g/L and 1.76 g/L/h, respectively. However, the lime treatment detoxification production process is not green enough, and there are problems such as the evaporator is easy to scale and produce a large amount of solid waste. Therefore, this paper attempts to use the hydrolysate without lime detoxification as the substrate to produce xylitol. However, the concentration and productivity of xylitol are not too high, and further research is still needed.
[*10] Yuan, X.; Tu, S.; Lin, J.; Shen, H.; Yang, L.; Wu, M. Combination of the CRP mutation and ptsG deletion in Escherichia coli to efficiently synthesize xylitol from corncob hydrolysates. Appl. Microb. Biot. 2020, 104, 2039-2050; DOI10.1007/s00253-019-10324-0.
Reviewer 3 Report
The article provided the efficient E. coli engineering produced xylitol using completely non-detoxified hemicellulose hydrolysate as the substrate to date. This study also demonstrated that alkali neutralization, high-temperature sterilization, and fermentation of the hydrolysate had important effects on the xylose loss rate and xylitol production. Overall, the manuscript is important for researchers in chemical production.
Listed are some comments regarding the submitted manuscript:
1. Line 25, Introduction part: all fonts of worlds should be unique
2. Line 28: What are the benefits of using an E. coli bacterium for chemical production?
3. Line 30, 34: E. coli → E. coli
4. Line 341: The conclusion part should summarize the optimal condition (pH, temperature, glucose concentration, sulfate ion concentration…) for xylitol production.
Author Response
- Line 25, Introduction part: all fonts of words should be unique
Response: Thank you for your constructive criticism. We carefully checked the font format of the introduction according to the submission template of this journal, and unified the font as Palatino Linotype and size 10.
- Line 28: What are the benefits of using an E. coli bacterium for chemical production?
Response: Thank you for your question. We have made corresponding supplements in the munuscript. Escherichia coli is one of the optimal host organisms for research related to metabolic engineering and synthetic biology because of its metabolic plasticity, its ability to assimilate both hexose and pentose sugars and grow rapidly under commonly used culture conditions, and the availability of substantial biochemical and physiological information and genetic engineering techniques.
- Line 30, 34: E. coli →E. coli
Response: We have made relative revisions.
- Line 341: The conclusion part should summarize the optimal condition (pH, temperature, glucose concentration, sulfate ion concentration…) for xylitol production.
Response: Thank you for your suggestions. We have made relative supplements.
Round 2
Reviewer 1 Report
The authors have revised the manuscript to address the questions from the first review.
Author Response
Thank you for your approval. We have carefully checked English language and style.
Reviewer 2 Report
The manuscript may be accepted but "special treatment" to minimize the impact of lignin must be included. Also, please include the lignin content in starting corn cob, resdiual solids, and hydrolysates.
Author Response
Response to referee's comments
The manuscript may be accepted but "special treatment" to minimize the impact of lignin must be included. Also, please include the lignin content in starting corn cob, resdiual solids, and hydrolysates.
Response: Thank you for your suggestions. We have added relative supplements in Part 3: Materials and Methods. The production method of corncob hydrolysate was described in the literature [*1]. The corncob powders were firstly treated with pectinase, and then reacted with dilute sulfuric acid (0.6%~1.0%) at 125 °C for 50 minutes. Hemicellulose was almost completely hydrolyzed to xylose, and cellulose and lignin remained unchanged as solid residues, which were then separated by a plate and frame filter. Therefore, no "special treatment" is needed to reduce the impact of lignin. The lignin content in starting corncob and acidolysis residue is about 20% and 35%, respectively. The hydrolysate contains almost no lignin.
[*1] Jiang, S.T.; Chen, D.S.; Li, M.; Liao, C.J.; Mao, B.X.; Zheng, Y.; Zhang, J.J.; Li, S.; Wang, X.X. The invention relates to a method for preparing xylose solution from corncob. Chinese patent. 2018, CN107893131A.